# YOLO-LFPD: A Lightweight Method for Strip Surface Defect Detection

**DOI:** 10.3390/biomimetics9100607

**Published:** 2024-10-08

**Authors:** Jianbo Lu, Mingrui Zhu, Kaixian Qin, Xiaoya Ma

**Affiliations:** 1Guangxi Key Lab of Human-Machine Interaction and Intelligent Decision, Nanning Normal University, Nanning 530001, China; lujianbo@nnnu.edu.cn; 2Guangxi Zhuang Autonomous Region Forestry Survey and Design Institute, Nanning 530011, China; zzz418229223@gmail.com; 3Department of Logistics Management and Engineering, Nanning Normal University, Nanning 530023, China; maxy519@126.com

**Keywords:** surface defect recognition, lightweighting, YOLO, pruning, FasterNet

## Abstract

Strip steel surface defect recognition research has important research significance in industrial production. Aiming at the problems of defect feature extraction, slow detection speed, and insufficient datasets, YOLOv5 is improved on the basis of YOLOv5, and the YOLO-LFPD (lightweight fine particle detection) model is proposed. By introducing the RepVGG (Re-param VGG) module, the robustness of the model is enhanced, and the expressive ability of the model is improved. FasterNet is used to replace the backbone network, which ensures accuracy and accelerates the inference speed, making the model more suitable for real-time monitoring. The use of pruning, a GA genetic algorithm with OTA loss function, further reduces the model size while better learning the strip steel defect feature information, thus improving the generalisation ability and accuracy of the model. The experimental results show that the introduction of the RepVGG module and the use of FasterNet can well improve the model performance, with a reduction of 48% in the number of parameters, a reduction of 13% in the number of GFLOPs, an inference time of 77% of the original, and an optimal accuracy compared with the network models in recent years. The experimental results on the NEU-DET dataset show that the accuracy of YOLO-LFPD is improved by 3% to 81.2%, which is better than other models, and provides new ideas and references for the lightweight strip steel surface defect detection scenarios and application deployment.

## 1. Introduction

Machine learning techniques are commonly employed in the detection of surface defects in steel. For instance, to enhance the usability of the original image, operations like denoising and enhancement are performed. Next, features such as texture and shape features are extracted to describe the feature information of steel surface defects. Finally, a classifier like support vector machines (SVMs) [1], random forests (Random Forests) [2], etc. is used to classify and recognize the extracted features. These methods can detect steel surface defects to some extent. However, they rely on hand-designed features and require a separate design of feature extractors and classifiers for defects of different types and scales. Therefore, they have limited generalisation capability. The emergence of deep learning methods in recent years has led to a significant breakthrough in steel surface defect detection technology. Deep learning methods can automatically learn feature representations in images with better generalisation ability and robustness by constructing deep neural network models. The convolutional neural network (CNN) [3] is widely used in steel surface defect detection due to its excellent performance in image processing. These methods are capable of extracting feature representations of steel surface defects effectively. They are also capable of end-to-end training and optimization, resulting in more accurate and efficient defect detection.

The current task of detecting defects in strip steel using deep learning models presents several challenges. These models typically require a large amount of labelled data for training, but the acquisition and labelling of strip steel defect images in practical applications are costly. As a result, there are limited training samples available, which makes it difficult to train the models. Secondly, strip steel defects come in various types and have complex morphology. Differentiating between these types of defects often requires a lightweight and highly efficient method of constructing a strip steel defect detection model based on deep learning. The deep learning model utilises feature learning and generalisation to improve the accuracy and efficiency of strip steel defect detection. This reduces manpower costs and promotes intelligent processes in strip steel production.

Currently, there are three mainstream strip defect detection algorithms: Mask R-CNN [4], EfficientDet [5], and YOLO (you only look once) [6]. The two-stage target detection algorithms typically involve two phases: candidate frame generation and frame classification. During the candidate frame generation phase, the algorithm extracts potential target regions from the input image using techniques such as the region proposal network (RPN). In the frame classification stage, the generated candidate frames undergo target category judgement through a classifier and are adjusted as necessary to obtain the final target detection result. Compared to the one-phase detection algorithm, the two-phase detection algorithm requires more computational resources and time to perform the two-phase operation, resulting in slower detection speed. This makes it unsuitable for application scenarios with high real-time requirements. Secondly, the generation of candidate boxes typically involves methods such as sliding windows or region proposal networks. These methods can result in significant computational and storage overheads when dealing with a large number of candidate boxes, which can limit the scalability of the algorithm. Furthermore, two-stage detection algorithms are not effective in detecting small targets due to the challenge of generating candidate frames efficiently in the candidate frame generation stage. This often leads to missed or false detections. As a result, one-stage detection algorithms, such as the YOLO algorithm, demonstrate better performance and generalisability in industrial applications.

YOLO (you only look once) [6] is a series of classical single-stage target detection algorithms designed to achieve fast and accurate target detection. The evolved versions of YOLO, including YOLOv5, YOLOX [7], YOLOv7 [8], and YOLOv8, employ a lightweight backbone network with high feature extraction and parameter efficiency. The model’s feature extraction process primarily consists of bottleneck structures. Terven, J.R. et al. [9] provide a comprehensive analysis of the development of YOLOv1 to YOLOv8. The authors argue that all official YOLO models from YOLOv5 onwards have fine-tuned the trade-off between speed and accuracy, with the aim of better adapting to specific applications and hardware requirements. There is no significant difference in accuracy from YOLOv5 to YOLOv8, and YOLOv5 has greater scalability and community resources than YOLOv8. During the testing of the NEU-DET dataset with YOLOv8, it was observed that the detection results for small targets were unsatisfactory and the training time was longer compared to YOLOv5. As a result, YOLOv5 was chosen over YOLOv8.

In order to achieve better results in strip defect detection with insufficient datasets, the YOLO-LFPD model is proposed in this paper. The Fasternet backbone network and RepVGG (Re-param VGG) are introduced on the basis of the YOLO-LFPD model with the aim of reducing redundant computations and memory accesses. RepVGG is a lightweight convolutional neural network structure with a simple but effective design, which aims to maintain high performance with a small number of parameters and low computational cost. In addition, the YOLO-LFPD model can be made lightweight by pruning techniques, which reduces the number of parameters and the amount of computation required. This greatly improves the inference speed of the model, reduces the deployment cost, and allows it to be used in resource-limited devices or environments while maintaining detection accuracy. The main contributions of this paper are as follows:In this paper, the pruning method is used to enable the model to select the appropriate parameter size for actual detection. When the parameter size is reduced to half of the original, the accuracy rate of mAP50 can still reach 75.6%. In addition, the replacement loss function and hyperparameter evolution methods can improve the accuracy of mAP50 by 6.5% when the model parameters are reduced.The RepVGG module simplifies the inference process while allowing the model to acquire more surface defect feature information, with a 0.8% increase in mAP50 accuracy, which reduces the depth of the model while being able to improve the extraction of surface defect image features from the strip. A better fusion effect was obtained by testing the location of the FasterNet module, with a 0.2% increase in mAP50 accuracy. In addition, the FasterNet module allows the model to ignore unimportant feature information, reducing the amount of computation while preventing model overfitting, and reducing the number of parameters and the amount of computation of the model, with the number of parameters being 63.6% of the original, and the GFLOPs being 73.8% of the original.Performance tests were conducted on the NEU dataset, GC10-DET dataset, and PKU-Market-PCB dataset. The experimental results demonstrate that the model’s accuracy and number of parameters in this paper are optimal compared to network models from the past two years. Additionally, the model’s ability to perform inference speed is also noteworthy, providing a reference for the application of surface defect detection of strip steel and the deployment of the model in actual production scenarios. The model serves as a reference for applying and deploying it in real production scenarios.

The article is structured into several sections. Section 2 compares the strip defect object detection model, the strip defect dataset, and the YOLO method. Section 3 describes some architectural details of the improved algorithm’s backbone network. Section 4 analyses the data experiments, and Section 5 and Section 6 conclude the article.

## 2. Related Work

### 2.1. Strip Defect Detection Technology

Traditional image processing methods are significant in detecting defects on steel surfaces. These methods employ image processing techniques to enhance image data and locate defect regions. Juan et al. [10] employed a histogram thresholding method that integrated empirical knowledge to detect residual oxide skins in stainless steel production. Neogi et al. [11] developed a global adaptive thresholding technique for defect detection in steel strips, which adjusts the threshold adaptively based on the number of pixels with grey values in the gradient image. Li et al. [12] and Chen et al. [13] used a region-growing algorithm for the detection of defects in steel plates with high accuracy.

Deep learning methods achieve end-to-end learning. This means that feature representation and target detection tasks are learned directly from raw data. In contrast, traditional methods require the manual design of feature extractors and classifiers, which can be a cumbersome and error-prone process. Furthermore, the deep learning model is highly adaptable and can effectively handle target detection tasks of varying scales, shapes, and poses. In contrast, traditional methods often require manual parameter adjustments and threshold settings.

Lu et al. [14] proposed a YOLO-FPD defect detection method, which includes spatial pyramid pooling fast advance (SPPF-A) and convolutional separated bottleneck (CSBL) methods, with an accuracy improvement of 1.8%. Liu et al. [15] proposed an MSC-DNet defect detection method with an inflated convolutional PADC and a feature enhancement module FESM, which enhances the extraction of multi-scale features to reduce confusing information. Zhao et al. [16] proposed an LSD-YOLOv5 defect detection method with a smaller bidirectional feature pyramid network (BiFPN-S), which improved the detection speed by 28.7% while increasing the accuracy by 2.4%. Zhao et al. [17] proposed a lightweight efficient detection network (LEDNet), which reduces the computational complexity while adaptively extracting the effective features. Zhou et al. [18] proposed an automated surface defect detection network, DPN-Detector, and proposed a residual space pyramid, RASPP, and a dual pyramid network, DPN. Wang et al. [19] proposed a new multi-scale feature fusion module, ATPF, based on which a deep learning network, CG-Net, was constructed for surface defect detection on strips. Chen et al. [20] proposed a fast surface defect detection network, DCAM-Net, for strips based on deformable convolution and attention mechanisms, and proposed a new enhanced deformable feature extraction block, EDE-block. Chen et al. [21] proposed an improved YOLOX method using variable focal length loss to solve the foreground-background class imbalance problem and predicted the IoU perceptual classification score as the detection ranking criterion. Li et al. [22] proposed a YOLOv5s-GC detection method by applying OTSU and a normal distribution enhancement algorithm (NDEA) to extract feature greyscale and introducing the GhostBottleneck module, which reduces the computation required by 48%.

The improvement of deep learning models in recent years is based on the transformer model and the YOLO model with improved convolutional neural networks. The main research direction in surface defect detection of strips is to improve the multi-scale spatial pyramid structure, the residual network structure, and the variant convolution. The reason for this is that the localisation of features in strip steel defect detection is inaccurate, and the size of the dataset restricts the depth of the model, which reduces the efficiency of the arithmetic power usage. Both the multi-scale spatial pyramid and the residual network are better at retaining the defective features and improving the accuracy of the recognition. Furthermore, the variant convolution reduces the wastage of the arithmetic power, which in turn reduces the size of the model and speeds up the detection efficiency.

### 2.2. Pruning

Pruning is a technique used to reduce the number of parameters and computational effort of a neural network model. The main idea is to improve the efficiency of the model by identifying and removing connections or parameters that contribute less to the performance of the model while maintaining its accuracy. Pruning techniques are usually classified into two types: structural pruning and weighted pruning.

Structural pruning, also known as network pruning, involves reducing the number of parameters in a model by removing entire neurons, channels, or layers. This process is typically performed after training by evaluating and filtering the model to identify the parts that can be pruned. Among the structural pruning methods for YOLO models, common ones include LAMP pruning [23], SLIM pruning [24], Group_slim pruning [25], Group_sl pruning [26], Growing_reg pruning [27], Group_taylor pruning [28], and others.

LAMP pruning (layer-adaptive magnitude-based pruning) is adaptive insofar as it considers the importance of each layer for pruning. This method is capable of reducing the amount of computation and number of parameters required while retaining the key feature extraction capability, which is highly suitable for target detection tasks such as YOLO with high real-time requirements. SLIM pruning (sparse learning with implicit magnitude pruning) can dynamically adjust the model complexity during the training process to adapt to the YOLO model hierarchical structure, thus significantly reducing the number of parameters and required computation under the premise of maintaining accuracy. Group_slim pruning is an effective method for reducing the redundancy of the YOLO model. It involves grouping pruning and retaining the important channels in each group while maintaining the overall structure of the model and the feature extraction capability. Group_sl pruning further optimises the Group_slim pruning strategy, which makes the pruned model more robust and efficient in practical applications. It is suitable for the deployment of YOLO models in different scenarios. Growing-reg pruning combines the advantages of pruning and regularisation, and can gradually adapt to the data distribution and task requirements during the training process. This enables the pruned YOLO model to reach a good balance between accuracy and efficiency. Group_taylor pruning employs Taylor expansion to estimate the relative importance of each channel, thereby enabling the retention of the most crucial feature channels for the YOLO model, which relies on channel features for target detection. This approach has been shown to significantly enhance the performance of the model following pruning.

Weight pruning is a technique used to reduce the number of parameters in a model by setting certain weights to zero. This method is typically performed during training and involves dynamically setting weights smaller than a certain threshold to zero. Common weight pruning methods include L1 pruning [29], which prunes by absolute magnitude, and pruning by gradient magnitude.

Pruning techniques offer several advantages, including reducing the number of model parameters and computations, improving model efficiency and lightweightness, reducing the risk of overfitting, and improving training and inference speed. Additionally, pruning can save storage space and energy consumption by effectively removing unnecessary parameters and connections in the model, thus reducing computational complexity and memory consumption. Pruned models are typically more lightweight and better suited for resource-constrained environments, such as mobile devices and embedded systems.

### 2.3. Hyperparameter Evolution and the OTA Loss Function

The YOLO target detection algorithm is an efficient method for detecting targets. However, its performance can be affected by the selection of hyperparameters. To optimize these hyperparameters, the genetic algorithm (GA) can be used as a heuristic search method. The GA is an optimization algorithm that involves a series of mathematical representations and operations. Firstly, a fitness function $f\left(x\right)$ is defined to evaluate the performance of an individual x. This is usually based on the model’s performance on a validation set. An individual x represents a hyperparameter combination, typically a vector or chromosome, containing multiple hyperparameter values. For example, $x=\left(x_{1},x_{2},\ldots ,x_{n}\right)$, where $x_{i}$ denotes the value of the ith hyperparameter. The initialised population P contains a collection of multiple individuals, e.g., $P=\left\{x_{1},x_{2},\ldots ,x_{m}\right\}$, where m denotes the size of the population.

During the selection operation, the parent individual is chosen based on its fitness value, $f\left(x\right)$. This can be achieved through methods such as roulette selection or bidding tournament selection. The crossover operation involves combining the genetic information of the parent individuals, $x_{1}$ and $x_{2}$. Finally, the mutation operation introduces a new combination of genes by randomly varying the genetic information of the individual x. For instance, an individual’s hyperparameters can be randomly perturbed or replaced. New offspring individuals are generated by selection, crossover, and mutation operations to update population P and obtain P’. The final hyperparameter combination is determined by the highest fitness when the maximum number of iterations is reached or the stopping condition is satisfied. The description above includes a mathematical representation of the fitness function $f\left(x\right)$, the individual representation x, the initialized population P, the selection operation $Select\left(P\right)$, the crossover operation $Crossover\left(x_{1},x_{2}\right)$, the mutation operation $Mutate\left(x\right)$, and the update of the population P’.

OTA (optimal transport assignment for object detection) is a loss function designed to solve the problem of target assignment in target detection. It adopts an ordering-based strategy to optimise performance by pairing the bounding box output from the detector with the real target. The main idea of OTA is to transform the target detection task into a Hungarian algorithmic problem to minimise the total loss of matching. OTA consists of two components: matching loss and no matching loss.

The matching loss evaluates the quality of matching by calculating the degree of match between the bounding box output by the detector and the real target. This process utilises the intersection over union (IoU) score as the matching metric. It determines the degree of overlap between the detected and actual boxes by calculating the ratio of the intersection area of the detected box and the actual box to their combined area. The OTA loss function then pairs the detection frames with the actual target using the Hungarian algorithm to minimise the total matching loss. The Hungarian algorithm, also referred to as the Hungarian method or the Kuhn–Munkres algorithm, is an optimisation algorithm used to solve the assignment problem. The time complexity of the Hungarian algorithm is $O\left(n^{3}\right)$, where n represents the number of objects and tasks. Equation (1) represents the matching loss, where N represents the number of detection frames, M represents the number of real targets, and $IoU_{i,j}$ represents the IoU value between the ith detection frame and the jth real target. The matching loss is shown in Equation (1)
(1)$$MatchingLoss=-\sum_{i=1}^{N}log\left(\frac{exp\left(-IoU_{i,j}\right)}{\sum_{k=1}^{M}exp\left(-IoU_{i,k}\right)}\right)$$

The no-match loss addresses the situation where real targets are not matched, ensuring that all detection frames receive appropriate penalties. The no-match loss can be calculated by determining the confidence scores of the unmatched detection frames using Equation (2), where λ represents the weight parameter and $IoU_{i,max}$ represents the maximum IoU value of the ith detection frame with respect to all real targets.
(2)$$aNo\;Match\;Loss=\lambda \sum_{i=1}^{N}{\left(1-IoU_{i,max}\right)}^{2}$$

The OTA loss function optimises the performance of the target detection model by minimising the total loss function, which combines matching and non-matching losses. The degree of match between the detection frame and the real target can be evaluated more accurately by introducing sorting and Hungarian algorithms.

### 2.4. Datasets

The NEU-DET dataset, the industrial metal surface defects GC10-DET dataset [30], and the circuit board surface defects PKU-Market-PCB dataset used in the experiments in this section are all from real production products, and their defects are closer to the actual situation. The images in the PKU-Market-PCB dataset are dominated by surface defects, which is suitable for the generalisation of the strip defects detection model as a test dataset. The circuit board surface defects dataset has been used in papers [31,32,33,34,35,36,37] on strip defect detection research in recent years.

The NEU-DET dataset is a public dataset created at Northeastern University for strip surface defect detection and contains six common defect types: roll marks, inclusions, scratches, indentations, pits, and cracks. The dataset features high-resolution images and accurate labelling of defect categories to provide a detailed picture of the subtle defect characteristics on the strip surface. As shown in Figure 1, the NEU-DET dataset totals 1800, where each class of defects contains 300 images with a size of 200 × 200 pixels. The training set, test set, and validation set will be randomly generated at 8:1:1 for the experiment.

Ten types of surface defects are included within the GC10-DET dataset as shown in Figure 2, i.e., Punch (Pu), Weld (Wl), Crescent Gap (Cg), Water Spot (WS), Oil Spot (Os), Silk Spot (Ss), Inclusions (In), Rolled Pit (Rp), Crease (Cr), and Waist Fold (Wf). The GC10-DET dataset consists of a total of 2294 images with an image size of 2048 × 1000 pixels. The experiment randomly generates the training set, test set, and validation set in 7:2:1, as shown in Figure 3.

The PKU-Market-PCB dataset has a total of six types of defects: leakage holes, rat bites, open circuits, short circuits, stray and stray copper. As shown in Figure 4. The experiment randomly generates the training set, test set, and validation set according to 7:2:1, and the number of defects of each type is shown in Table 1, totalling 693 images.

## 3. Modelling Design

YOLO-FPD [14] is one of our models for strip defect detection, and our proposed residual module LSTM as well as the spatial pyramid pooling module SPPF-A have achieved good results, with a 1.7% improvement in accuracy compared to the original YOLOv5 model. To cope with the possible inadequacy of the dataset images, we propose the SSIN image enhancement method, which varies the original NEU-DET dataset images and improves the accuracy by 37.5% while keeping other conditions constant. However, the YOLO-FPD model size does not decrease significantly, and there is room for improvement in lightweighting. In addition, when the SSIN image enhancement method is not used and the dataset images are not sufficient, it is prone to training overfitting, which affects the accuracy rate. Therefore, we propose the YOLO-LFPD model, which mainly introduces the Fasternet backbone network, RepVGG (Re-param VGG), and pruning techniques to maintain high performance while having a small number of parameters and low computational cost, providing a feasible application of the model in the resource-limited devices or environments.

The YOLO-FPD [14] model mainly includes the neck network, backbone network, and ASFF detector head. In_channel denotes the number of input channels, out_channel denotes the number of output channels, kernel_size denotes the size of the convolution kernel, and stride denotes the step size. In a backbone network, CSBL greatly reduces the number of parameters and computations of the model by decomposing the standard convolution operation into two independent operations: depth convolution and point-by-point convolution. This helps to reduce the size of the model and improves the lightweight and efficient performance of the model. With the introduction of CBAM in the neck network, a CBAM module is added at the end of each CSBL module, and a CBAM module is also added after the SPPF-A layer of the neck network.

In order to improve lightweightness, Fasternet and RepVGG structures are introduced and the YOLO-LFPD model is proposed. As shown in Figure 5, Fasternet, as a lightweight feature fusion network with small model parameters and computational complexity, is able to effectively extract the feature information of the strip image and enhance the model’s ability to detect defects at different scales through multi-scale feature fusion. RepVGG, as a simple and efficient convolutional neural network structure with good model compression effect and versatility, can effectively reduce the computational burden of the model and maintain high detection accuracy. Taken together, combining Fasternet and RepVGG into the YOLO-LFPD model can improve the lightweightness and high efficiency of the model while maintaining the superior performance of the model on the defect detection dataset. In addition, the pruning technique can reduce the number of parameters and the computation requirements of the model and improve the inference speed. The combined use of these techniques can effectively improve the performance and efficiency of the model in the strip steel surface defect detection task.

### 3.1. FasterNet Network

FasterNet[38] is a new family of neural networks designed to achieve high speed and high efficiency in various visual tasks, and its efficient architecture is capable of processing large amounts of image data at high speeds in strip production lines to meet the requirements of high frame rate and low latency, thus improving productivity and quality control. It employs a novel partial convolution operator called PConv (partial convolution) and the off-the-shelf PWConv (pointwise convolution) as the main construction operators to reduce the computational redundancy and the number of memory accesses, thus improving the speed and efficiency of the network. When the input and output feature maps have the same number of channels, PConv has FLOPs of $h\times w\times k^{2}\times c_{p}^{2}$. If we take the default value of $c_{p}=c/4$ for the $c_{p}$ channel, the FLOPs are 1/16 of those of regular Conv. The memory access of PConv is approximately $h\times w\times c_{p}+k^{2}\times c_{p}^{2}\approx h\times w\times {2c}_{p}$, if we take the default value of $c_{p}=c/4$, its memory access is only 1/4 of that of conventional Conv.

To fully and efficiently utilize the information from all channels, point-by-point convolution (PWConv) is further appended to PConv as shown in Figure 6. Their effective receptive field on the input feature map looks like a T-shaped Conv, which focuses more on the centre position than the regular Conv of a uniformly processed patch. In terms of small target detection, PConv can flexibly deal with localized regions of the feature map to enhance the recognition of small defects on the strip surface, and further improve the detection accuracy by combining with multi-scale feature extraction methods.

Different locations of FasterNet integration into the model will have different impacts on the feature extraction, fusion, and final detection performance of the model. Introducing the backbone network can make early feature extraction efficient and flexible and improve the quality of the initial feature map; introducing the neck network can improve the detection accuracy of both small and large targets; and introducing the head detection network can utilize more efficient feature representations when predicting the category and location. Through the 4.4 Fusion Location experimental test, this paper chooses to introduce FasterNet into the backbone network.

Figure 7a shows the original YOLO structure and Figure 7b shows the YOLO-FPD model structure after replacing the original C3 module with the C3Faster module. The FasterNet backbone network is adopted to accelerate the processing speed of target detection, enabling the system to respond to input images faster and achieve higher frame rates in real-time scenes. Furthermore, FasterNet boasts a low computational complexity and requires fewer memory accesses, making it ideal for efficient inference in resource-constrained environments, such as embedded systems and mobile devices.

The DCNv2 module in YOLO-FPD was replaced with normal convolution to avoid conflicts in strategy when using the two variants of convolutional structures together. If the DCNv2 module is used, overfitting may occur, resulting in a model that performs well on the training set but poorly on the validation and test sets. It is important to note that each convolutional variant has its own specific feature extraction method and optimization goals. Combining different structures can lead to conflicts or incoherence in feature expression, which can prevent the model from effectively learning useful features or even interfere with each other, reducing the model’s performance. Additionally, different convolutional structures may require distinct training strategies, regularization methods, or optimization algorithms to achieve optimal results. When combining them, the original optimization strategy may no longer be applicable, and the training process needs to be carefully adjusted and optimized.

### 3.2. RepVGG

RepVGG is a lightweight convolutional neural network architecture based on the repetitive module design idea. It aims to achieve efficient model training and inference. The network was proposed by a team of researchers at Microsoft Research Asia in 2021 as an innovative solution that departs from the traditional convolutional neural network design paradigm, offering higher efficiency and greater versatility. Convolutional neural networks (CNNs) typically comprise convolutional, pooling, and fully connected layers. These structures contain a large number of parameters, leading to high model complexity, slow training, and low inference efficiency. RepVGG takes a concise and efficient approach by dividing the network into RepVGG blocks, each containing simple convolutional layers, BatchNormalization layers, and ReLU activation functions. The network is built by stacking these blocks. The structure of RepVGG modules is illustrated in Figure 8.

The RepVGG block’s core components consist of normal convolutional layers with a fixed kernel size of 3 × 3. By adjusting parameters such as the number of output channels and step size, different sensory fields and feature extraction capabilities can be achieved. The BatchNormalization layer accelerates the model’s convergence process and improves its stability. The ReLU activation function introduces nonlinearities that enhance the network’s expressive power. By stacking RepVGG blocks repeatedly, RepVGG networks with varying depths and complexities can be constructed. This feature enables RepVGG to operate in environments with limited computational resources and adapt to various tasks and scenarios. RepVGG’s model training and inference are highly efficient. Another notable feature of RepVGG is its ability to maintain high efficiency in model training and inference. Due to its simple network structure and small number of parameters, RepVGG exhibits fast convergence during training and achieves superior model performance quickly. Additionally, RepVGG has low computational complexity and memory usage during inference, making it ideal for resource-constrained environments like mobile devices, embedded systems, and edge computing.

Figure 9a shows the original YOLO structure and Figure 9b shows the model structure of the YOLO-FPD with the integrated RepVGG module. RepVGG simplifies the computational complexity by using numerous 3 × 3 convolutional layers and eliminating the intricate branching structure. This design concept accelerates the forward inference process of the model, making it suitable for real-time detection scenarios. The language used is clear, objective, and value-neutral, and the technical terms are consistent throughout the text. The grammar, spelling, and punctuation are correct. No changes in content have been made. This not only ensures objectivity but also reduces the complexity of model maintenance and debugging. It is crucial to maintain a clear and concise writing style in technical writing. RepVGG’s structure allows for easy conversion to a more efficient inference structure through reparameterisation techniques after training. This flexibility enables the full exploitation of RepVGG’s structural advantages during the training phase while optimizing the model’s performance and efficiency during deployment. Furthermore, RepVGG’s improved feature extraction capability positively impacts YOLO-FPD’s ability to detect objects of varying scales.

## 4. Experiments and Analyses

This section evaluates the performance of YOLO-FPD in detecting strip surface defects after implementing FasterNet and RepVGG.

The performance of these models is compared to other models. The experiment sets the epoch of each model to 400, with the initial learning rate set to 0.1 and then gradually converging to 0. The 400 epoch ensures that all models involved in the comparison are trained to the maximum extent possible. The optimizer uses SGD. The GC10-DET dataset has a high image resolution, but due to limitations in the experimental device’s video memory, the training batch size is set to 16 for both training and testing. The NEU dataset and the PKU-Market-PCB dataset have a training batch size of 32. Please refer to Table 2 for details on the experimental environment.

The metrics used in this paper include mAP (mean accuracy), frames per second (FPS), number of model parameters (params), number of floating-point computations (GFLOPs), size of pre-trained models (MB), and number of layers of the YOLO model (LAYERS) in the COCO metric.

mAP (mean average precision) is a commonly used evaluation metric in target detection tasks, providing a comprehensive measure of model performance by calculating and averaging the average precision (AP) values for each type of target under different IoU (intersection-parity ratio) thresholds. mAP is calculated by integrating the precision-recall (P-R) curve, while mAP combines the AP values for a number of IoU thresholds (from 0.5 to 0.95) of AP values, reflecting the combined performance of the model at different confidence levels and thresholds. mAP@0.5 is a commonly used simplified metric that is particularly suitable for defect detection tasks.

FPS is used to evaluate the real-time processing capability of the model, i.e., the number of image frames that can be processed per second. The number of model parameters (params) indicates the total number of trainable parameters in the model, reflecting the complexity of the model. The amount of floating point computations (GFLOPs) is used to measure the amount of computation required for each inference of the model. Pre-trained model size (MB) refers to the size of the model on the storage medium and affects the ease of deployment. The number of layers (LAYERS) of the YOLO model indicates the depth of the model and affects the capability and complexity of feature extraction.

### 4.1. Dataset Experiments

In this study, the novel model named YOLO-LFPD is proposed. The YOLO-LFPD model significantly improves the inference speed by introducing lightweight network modules such as RepVGG and FastNet while maintaining high detection speed and accuracy. Among them, the YOLO model has five different sizes and complexities from small to large: n, s (small), m (medium), l (large), and x (extra large), with the difference only in the number of channels, where n is optimized for Nano devices (e.g., NVIDIA Jetson Nano), which is the fastest, although it sacrifices accuracy. YOLO-LFPD uses a size class of s, so YOLOv5s and YOLOv7s were chosen for comparison. The epoch is set to 400, the learning rate is initially 0.1 and then decreases to 0. The optimizer uses SGD and the batch size is set to 64.

As shown in Figure 10, compared with traditional models such as YOLOv5, YOLOv7, and Faster R-CNN, our model exhibits better performance when the dataset image resources are not sufficient to meet the training requirements. Although the accuracy of YOLOv5 improves rapidly in the early stages of training, the accuracy cannot be improved further due to the overfitting situation of the model, peaking at nearly 400 epochs. YOLO-LFPD, on the other hand, reduces the depth of the model and increases the residual structure compared to YOLO-FPD, which allows better attention to defective features with an insufficient number of samples, and peaks when training reaches about 250 iterations, which suggests that the optimised structure captures the target features better.

The training loss and validation loss for each model are shown in Figure 11a. The experimental results can be seen from the loss function curves in Figure 11b, although YOLOv5 has a lower loss value during training compared to the other models, the value starts to rise at about 100 epochs of the validation loss, which is already overfitting, and therefore, the network accuracy starts to decrease, while the YOLO-LFPD loss curve is more stable due to the addition of more residual structure loss curves. It can be seen that when training the model using a dataset with an insufficient number of images, once overfitting occurs, it is difficult to continue to improve the accuracy.

In order to verify the enhancement of the YOLO-FPD model in target detection results, several classical algorithms for strip steel surface defect detection are selected, including FasterRCNN, SSD, Efficiendet, YOLOv5, and YOLOv8. In addition, there are also improved models for steel surface defect detection carried out in the last two years, such as WFRE-YOLOv8s, ST-YOLO, LFF-YOLO, etc., and the results are shown in Table 3, in which the number of channel parameters selected by YOLO-LFPD-n is the same as that of YOLOv5-n, which is a more lightweight version that reduces the number and size of model parameters. It can be seen that the two improved models, YOLO-LFPD and YOLO-FPD, excel in performance metrics. The YOLO-LFPD model achieves 81.2% in terms of accuracy, while YOLO-FPD also achieves a satisfactory 79.6% accuracy. In addition, both improved models perform well in terms of the number of parameters and the number of floating-point computations, which are 6.4 M parameters and 14.1 GFLOPs and 13.2 M parameters and 20.2 GFLOPs, respectively. In contrast, other models may be a little less accurate or higher in terms of the number of parameters and computations, and the WFRE-YOLOv8s, although it achieves a single accuracy of 93.8% in Pa, the accuracy in Ps is too low; ST-YOLO reaches 97% in Sc, but the number of parameters of the model is too large compared with other one-stage models. Compared with several mainstream and improved algorithms, the results of YOLO-LFPD are competitive. Compared with the YOLOv5 baseline, the accuracy is improved by 3.9%, the GFLOPs on the model size is 87% of the original, and the params is 52% of the original. It can be clearly seen that the two improved models, YOLO-LFPD and YOLO-FPD, achieve better performance on the NEU dataset, with both higher accuracy and relatively lower parameter counts and number of computations.

The experimental results on the GC10-DET dataset are shown in Figure 12. In addition, since the accuracy of the model fluctuates a lot during the training process, in order to test whether there is any effect of different parameters on the model, it is adjusted in three directions, namely, increasing the confidence threshold, increasing the number of batches, and increasing the depth of the model, in which YOLO-LFPD-A, YOLO-LFPD-B, and YOLO-LFPD-C correspond to the above directions in turn.

Table 4 shows the accuracy of the GC10-DET dataset. YOLO-LFPD achieves 72.8% accuracy, which is a significant improvement over YOLOv5s’ 65.5%. LSD-YOLOv5 achieves 67.9% accuracy, slightly higher than YOLOv5, with parameters of 2.7 M and 9.1 GFLOPs, giving it lightweight and efficient advantages. In the recent one-year model, MFAM-Net accuracy is 66.7%, and YOLOX accuracy has improved to 70.5%. In comparison, YOLO-LFPD outperforms LSD-YOLOv5 in terms of accuracy, despite having a higher number of parameters and floating-point computations. WFRE-YOLOv8s achieves a higher accuracy of 69.4% but at the cost of a higher number of parameters. Overall, YOLO-LFPD has the best performance.

The results on the PKU-Market-PCB dataset are shown in Table 5, where the accuracy of YOLO-LFPD reaches 98.2%, which is a significant improvement from 94.7% in YOLOv5s. Among the models in recent years, TDD-Net achieves an accuracy of 95.1%, YOLO-MBBi achieves an accuracy of 95.3%, and DInPNet achieves an accuracy of 95.5%. PCB-YOLO has an accuracy of 96.0%, which is higher than that of the comparative models in recent years, while YOLO-LFPD has an accuracy of 98.2, thus YOLO-LFPD has a better overall performance.

### 4.2. Ablation Experiment

The ablation experiment environment configuration is consistent with the above experiments, and this experiment constructs the YOLO-LFPD model by replacing the FasterNet backbone network and adding the RepVGG module to examine its effect on the model recognition accuracy. The experimental results are shown in Table 6, replacing the FasterNet backbone network or adding the RepVGG module can improve the accuracy of the YOLO-LFPD model, and there is not much difference between the two improved accuracies. In terms of the number of parameters, the FasterNet backbone network uses the Frobenius paradigm for simple convolution to reduce the complexity of the model, and the number of parameters is reduced by 4.8 M, and the GFLOPs are reduced by 5.3; the RepVGG module, with its single-path architecture of high flexibility and parallelism, saves a lot of memory in the computation, and the number of parameters is reduced by 2 M, and the GFLOPs are reduced by 0.4 M. The RepVGG module’s reparameterisation can make the model structure more compact and at the same time have a higher accuracy, and FasterNet can make the model ignore the feature information which is not very useful, when RepVGG and FasterNet are used together, the model’s recognition accuracy is greatly improved, the accuracy is 81.2%, and the number of params is reduced to 5.3; the single path architecture of the RepVGG module is flexible and parallel, which saves a lot of memory when computing, the number of params is reduced by 2 M, and the GFLOPs are reduced by 0.8. The params parameter count is 48.5% of the original and the GFLOPs are 69.8% of the original.

### 4.3. Experiments on Reasoning Speed

The experiments on inference speed were conducted using the NEU dataset. The model’s recognition speed for each image was compared. YOLO-LFPD’s FasterNet backbone network is designed efficiently, resulting in a significant increase in the speed of the feature fusion and up-sampling modules of the network. This enables the model to perform better on real-time target detection and edge devices, meeting real-time requirements while reducing the computational burden on edge devices. During the training phase, YOLO-LFPD has a faster training speed compared to the original YOLO, allowing for quicker model training. Table 7 shows that YOLO-LFPD has an inference speed of 5.4 ms per image, which is a 22.9% improvement in speed compared to YOLOv5s. Additionally, the amount of parameters is reduced by 47% and the GFLOPs are reduced by 12%. YOLO-LFPD-n reduces the inference time by 16% compared to YOLOv5n’s single image inference time. Additionally, the YOLO-LFPD model has a faster inference speed compared to YOLOv8s and YOLOv7-Tiny. The results demonstrate that the YOLO-LFPD model achieves outstanding performance in terms of model recognition accuracy and overall optimization.

### 4.4. Fusion Position Experiments

In this paper, the FasterNet module is introduced into the backbone network, the feature pyramid, and the detection head of the YOLO-LFPD model, respectively, and experimental comparisons are conducted on the NEU dataset. The experimental epoch is set to 400, the learning rate has an initial value of 0.1 and then decreases to 0, the optimizer uses SGD, and the batch size is set to 64. FasterNet is able to make full use of its highly efficient feature extraction capability, which significantly improves the quality of the feature maps. As shown in Table 8, the model of the backbone network fused with FasterNet has significantly improved in both mAP@0.5 and FPS, achieving higher detection accuracy and real-time performance.

### 4.5. Parametric Experiments

The experiment used YOLO-LFPD-n and YOLO-LFPD as test models, NEU as the test dataset, and trained for 400 epochs with a batch size of 32 to test the effect of the GA genetic algorithm and OTA loss function. The results are presented in Figure 13. Figure 13a shows that using both the GA genetic algorithm and OTA loss function improves the model’s accuracy and convergence speed. The model’s performance is largely affected by hyper-parameters such as grid size, number of a priori frames, and weight of the loss function. Optimising these parameters with the genetic algorithm can find more suitable settings for the task. The OTA loss function can improve the accuracy of target detection by more accurately matching the detection frame with the real target. Additionally, the global optimization strategy of OTA is more sensitive to the problem of sparse target detection, allowing for better handling of missed or multiple detections and resulting in improved accuracy and robustness of detection. The YOLO-LFPD-n model’s accuracy is enhanced by the combined use of the GA genetic algorithm and the OTA loss function. This suggests that the two methods produce equally good results when used together. Figure 13b shows that while the GA genetic algorithm and the OTA loss function do not significantly improve the accuracy of the YOLO-LFPD model, they do improve the model’s stability during the training process.

Table 9 shows the experimental results. The YOLO-LFPD-n model, using the OTA loss function, improved mAP50 accuracy by 3.6% and mAP95 by 1.4%. The GA genetic algorithm improved mAP50 accuracy by 3.4% and mAP95 by 1.9%. The combined use of both methods resulted in an improvement of mAP50 accuracy by 4.3% and mAP95 by 2.1%, which is a significant improvement. For the YOLO-LFPD model, using both methods, mAP50 accuracy decreased by 0.7% and mAP95 accuracy improved by 0.1%. Based on the experimental data, it is suggested that the smaller YOLO-LFPD-n model can still learn effectively even when the dataset sample size is insufficient, resulting in improved accuracy. Although YOLO-LFPD is prone to overfitting, the OTA loss function and GA genetic algorithm can improve the extraction of feature information of defects during training. However, this improvement leads to overfitting during validation, which can be prevented by either expanding the dataset or reducing the model size.

### 4.6. Experiment on Pruning

To examine the impact of various pruning methods on the model, this section conducts pruning experiments on the NEU dataset using a batch size of 32. During the experiments, it was discovered that the model’s convergence was slower after pruning, resulting in an epoch change to 500. The experiments on pruning were carried out using a pre-trained YOLO-LFPD model with 400 epochs. The pruning methods used were LAMP pruning [23], SLIM pruning [24], Group_slim pruning [25], Growing_reg pruning [27], Group_taylor pruning [28], and L1 pruning [29]. The compression multiplication rate of pruning is represented by ×. 2.0× is compressed to 50% of the original. The pruning strategy involves disregarding the final layer of the FasterNet backbone network module, the ASFF module, and the CARAFE module. Algorithm 1 code is as follows:
**Algorithm 1** Pruning Judgment Skip Code1: for k, m in model.named_modules():2:   if isinstance(m, ASFF_Detect):3:     ignored_layers.append(m)4:   if isinstance(m, Faster_Block):5:     ignored_layers.append(m.mlp[−1])6:   if isinstance(m, CARAFE):7:     ignored_layers.append(m)

Table 10 shows the pruning effect, with L1 representing weight pruning, Group_norm [50] representing a normalization method, and the rest representing structural pruning. The table indicates that the overall effect of structural pruning is not ideal, with SLIM being the worst performer, achieving only 68.5% accuracy, while Group_taylor achieves 73.1% accuracy. On the other hand, L1 pruning and Group_norm achieve good results at compression rates around 1.5×.

### 4.7. Comparison of Forecast Frames

To better comprehend the impact of the improved YOLO-LFPD module on actual detection, we will compare the prediction frames of YOLO-LFPD and YOLO-FPD.

Figure 14 shows that YOLO-LFPD is capable of accurately detecting and classifying a wide range of irregular features, even when they overlap. In contrast, YOLO-FPD struggles to detect defects in a region as a whole due to insufficient training samples, resulting in prediction boxes with large gaps at the boundaries.

As can be seen from Figure 15, YOLO-LFPD possesses a sharper judgement, and in the face of the small and dense defects in Figure 15d, it is still able to distinguish the boundaries of their shapes more clearly, whereas YOLOv5 suffers from more miss-detection and misdetection.

### 4.8. Heat Map Comparison

To facilitate a more intuitive understanding of the improved module of YOLO-LFPD on actual detection, we will compare the prediction frames of YOLOv5 and YOLO-LFPD.

Figure 16 shows that the heat map of YOLO-LFPD has a stronger response in the target region. A comparison of the pixel value distribution of the heat maps generated by the two models reveals that the heat map of YOLO-LFPD is brighter and more concentrated in the target region, indicating a stronger target detection response. This suggests that the model can identify the target region more accurately and with greater confidence.

## 5. Conclusions

In this paper, based on the YOLO-FPD model, the YOLO-LFPD model is improved by introducing the RepVGG module and the FasterNet backbone network to get better performance in strip surface defect detection. RepVGG has a simple and efficient network design, adopting the idea of reparameterisation, combining convolutional layers with residual structures, which makes the whole network more compact and easier to train. This design greatly reduces the number of parameters and computational complexity of the model, enabling the YOLO model to have faster inference speed and lower memory consumption while maintaining high detection accuracy. FasterNet adopts a series of innovative operators, such as partial convolution (PConv), which improves the efficiency of the model in extracting spatial features; by reducing redundant computations and memory accesses, FasterNet is able to significantly increase the running speed of the model while maintaining a high accuracy rate. Experimental results on the strip steel surface defects dataset with an insufficient or unbalanced number of images show that the inference time for a single image is reduced by 22.9%, the amount of parameters is reduced by 47%, and the GFLOPs are reduced by 12%, which indicates that the improved YOLO-LFPD model has excellent comprehensive performance and exceeds the original model YOLOv5s in inference speed. This paper carries out pruning, substitution loss function, and hyperparameter evolution experiments in three aspects, and the experimental results show that substitution loss function and hyperparameter evolution experiments have better results on the YOLO-LFPD-n model with a smaller number of parameters, and the reasonable confidence threshold and model depth avoid the model from being overfitted, so as to improve the accuracy of the model, and these experiments fully prove the reasonableness of the YOLO-LFPD model. The main research results of this paper are summarized as follows:Pruning enables the model to choose the right parameter size for the actual detection, and the mAP50 accuracy still reaches 75.6% when the number of parameters is reduced to half of the original. In addition, the substitution loss function and hyperparameter evolution methods provide a better improvement in accuracy with smaller model parameters, and the mAP50 accuracy is improved by 6.5%.The RepVGG module simplifies the inference process while allowing the model to acquire more surface defect feature information, with a 0.8% increase in mAP50 accuracy, which reduces the depth of the model while being able to improve the extraction of surface defect image features from the strip. A better fusion effect was obtained by testing the location of the FasterNet module, with a 0.2% increase in mAP50 accuracy. In addition, the FasterNet module allows the model to ignore unimportant feature information, reducing the amount of computation while preventing model overfitting, and reducing the number of parameters and the amount of computation of the model, with the number of parameters being 63.6% of the original, and the GFLOPs being 73.8% of the original.Performance tests were conducted on the NEU dataset, GC10-DET dataset, and PKU-Market-PCB dataset. The experimental results demonstrate that the model’s accuracy and number of parameters in this paper are optimal compared to network models from the past two years. Additionally, the model’s ability to perform inference speed is also noteworthy, providing a reference for the application of surface defect detection of strip steel and the deployment of the model in actual production scenarios. The model serves as a reference for applying and deploying it in real production scenarios.

## 6. Prospect

This paper investigates the existing deep convolutional network model structure, optimises and improves it, and achieves relevant results. However, there are still many shortcomings in the treatment.This paper tests the performance of the improved model using the NEU dataset, the GC10-DET dataset, and the PKU-Market-PCB dataset. However, it is important to note that the data images used suffer from an imbalance in the number of labelled types, inaccurate labelling, and omission of labels. Subsequently, it is recommended to utilise GAN and other adversarial networks for image generation to improve and verify the model’s generalisation ability. This will enable the model to detect a wider range of steel surface defects and expand its potential applications.This paper focuses solely on detecting surface defects in strip steel and does not extend to other industrial production areas. Future work should explore target detection of surface defects in other workpieces and semantic segmentation of images to enrich research results in the field of steel surface defect detection.The lightweight network models proposed in this paper have only been tested on public datasets and have not yet been deployed to mobile or edge devices for real-world performance testing. Future work should focus on testing and validating the models’ performance in real-world scenarios of strip steel surface defect detection after deployment to devices.

## Figures and Tables

**Figure 1 biomimetics-09-00607-f001:**
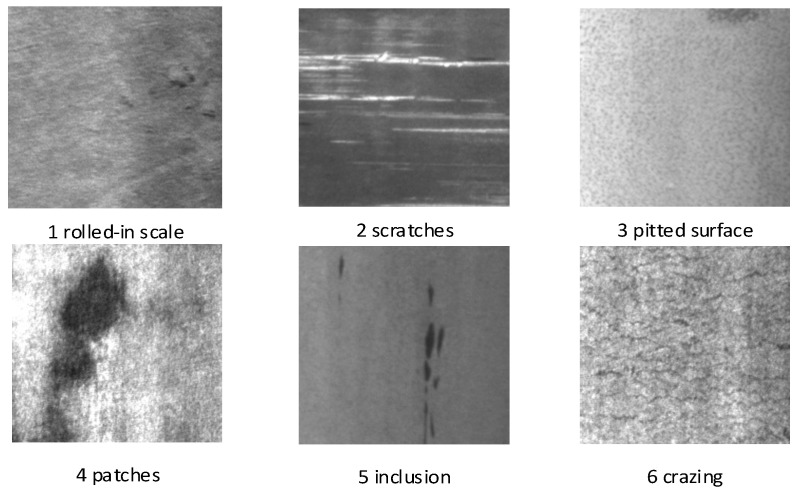
NEU surface defect dataset.

**Figure 2 biomimetics-09-00607-f002:**
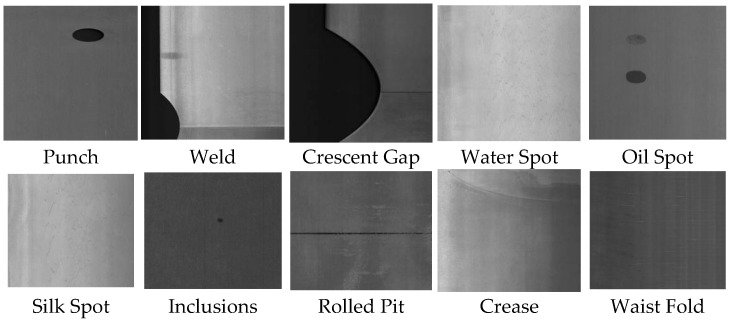
GC10-DET surface defect dataset.

**Figure 3 biomimetics-09-00607-f003:**
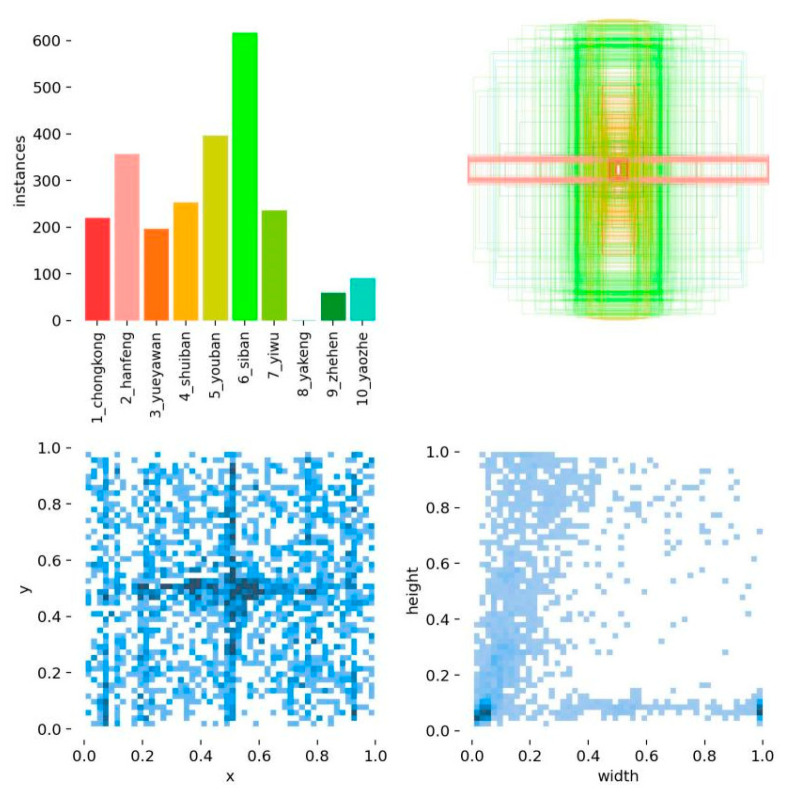
Image distribution of GC10-DET surface defect dataset.

**Figure 4 biomimetics-09-00607-f004:**
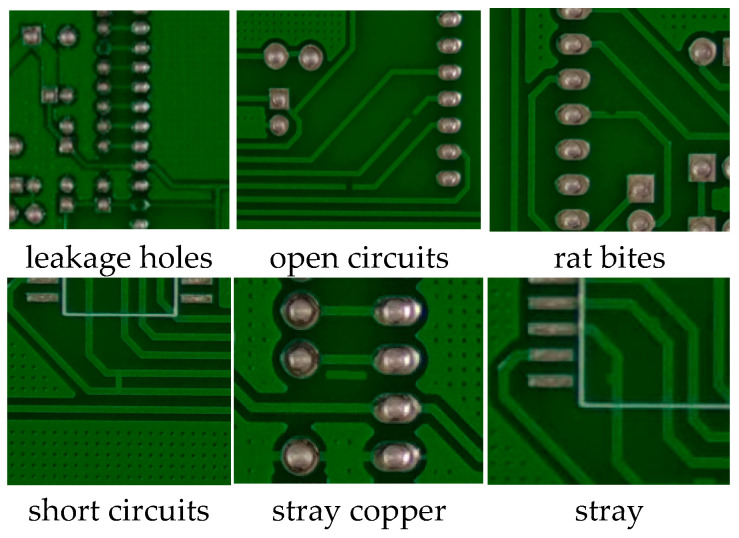
PKU-Market-PCB surface defect dataset.

**Figure 5 biomimetics-09-00607-f005:**
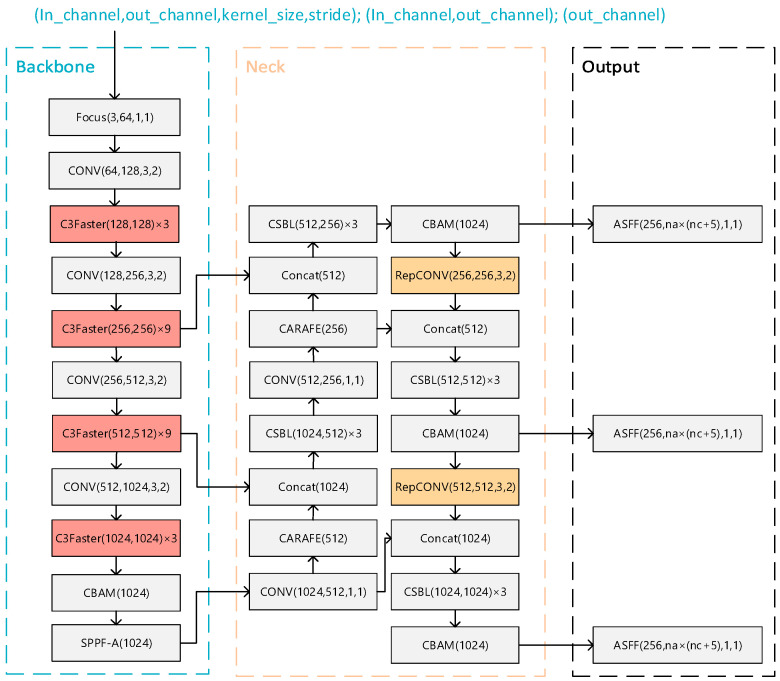
YOLO-LFPD structure.

**Figure 6 biomimetics-09-00607-f006:**
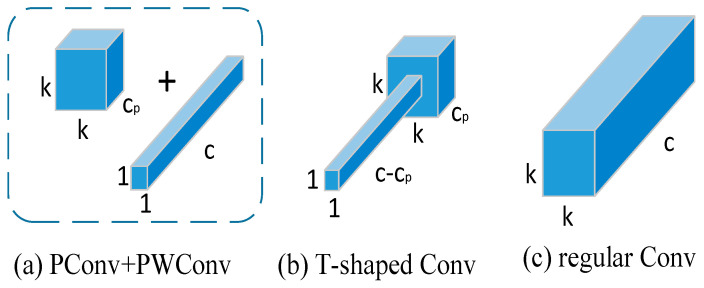
Comparison of convolutional variants.

**Figure 7 biomimetics-09-00607-f007:**
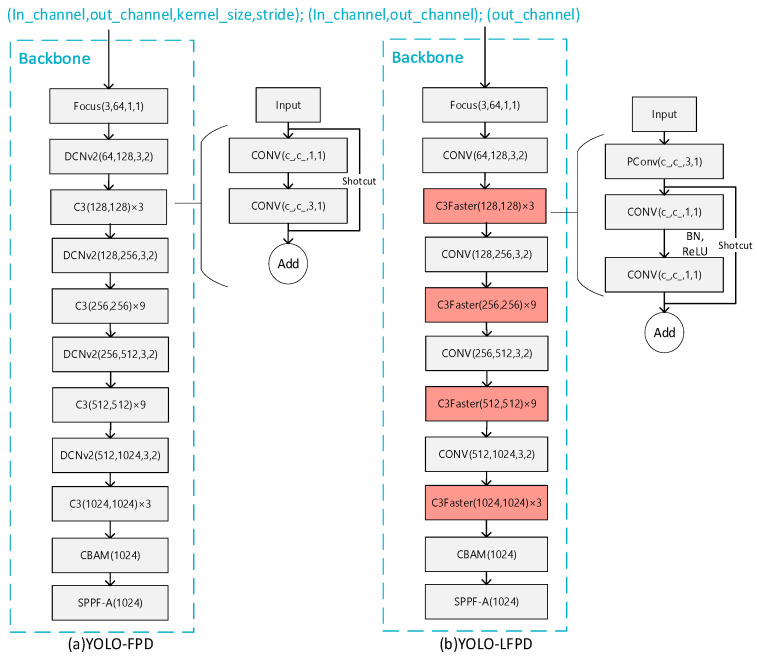
(**a**) for the original structure (**b**) for the FasterNet replacement location.

**Figure 8 biomimetics-09-00607-f008:**
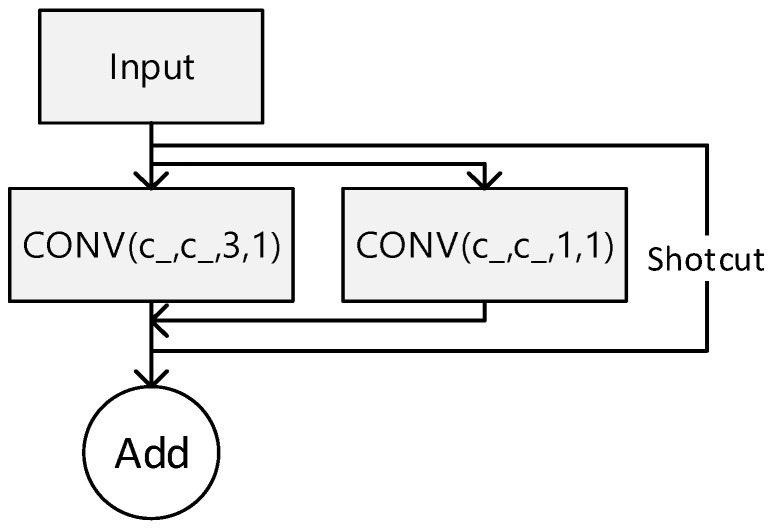
RepVGG.

**Figure 9 biomimetics-09-00607-f009:**
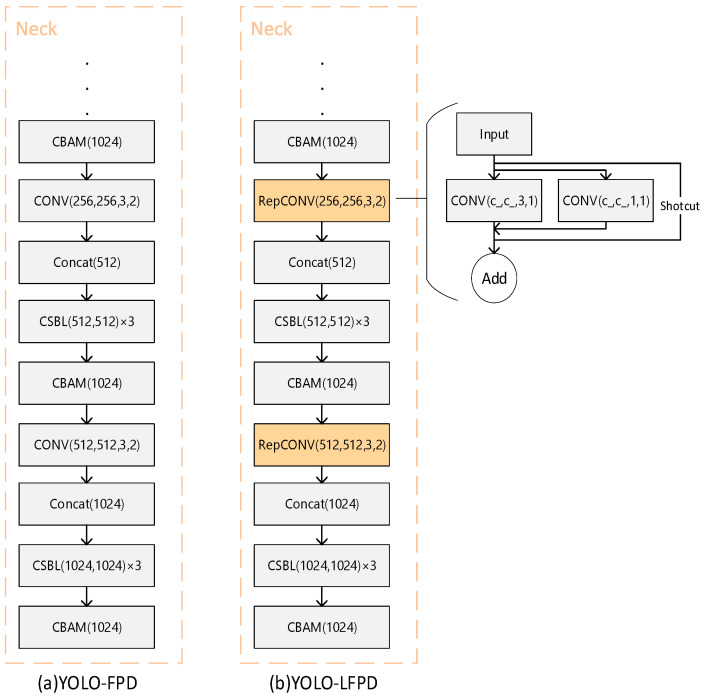
(**a**) for the original structure (**b**) for the RepCONV replacement position.

**Figure 10 biomimetics-09-00607-f010:**
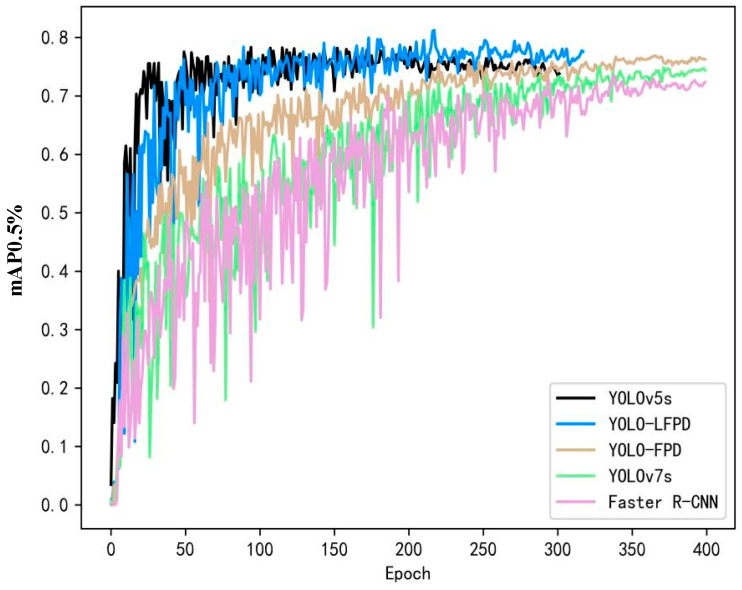
Line graph displaying the experimental results of YOLO-LFPD on the NEU dataset.

**Figure 11 biomimetics-09-00607-f011:**
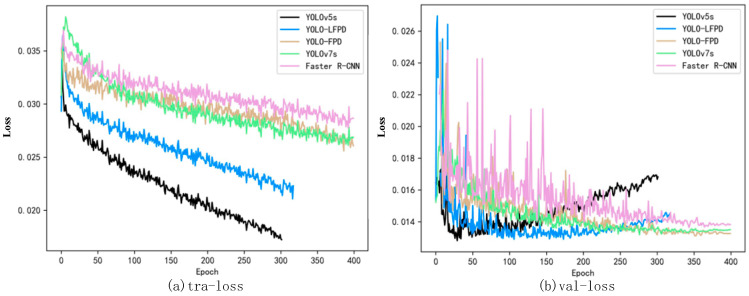
The YOLO-LFPD loss function was applied to the NEU dataset, with (**a**) representing the training loss and (**b**) representing the validation loss.

**Figure 12 biomimetics-09-00607-f012:**
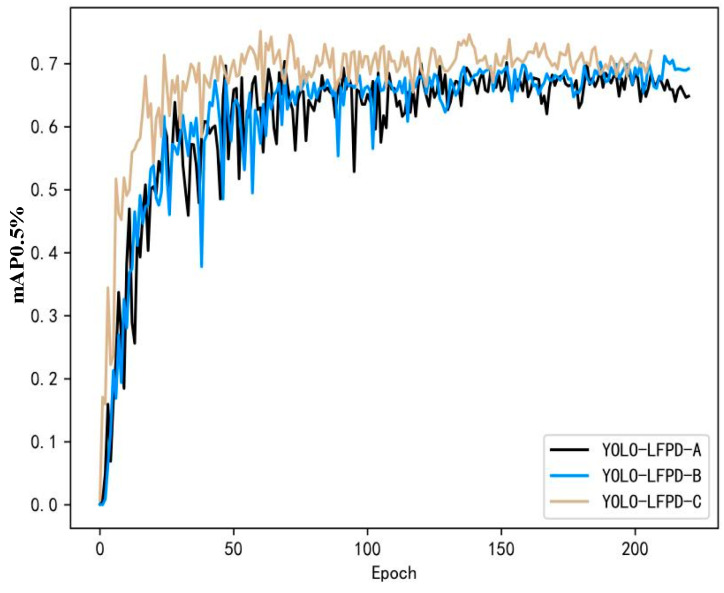
YOLO-LFPD accuracy fold plot on GC10-DET dataset.

**Figure 13 biomimetics-09-00607-f013:**
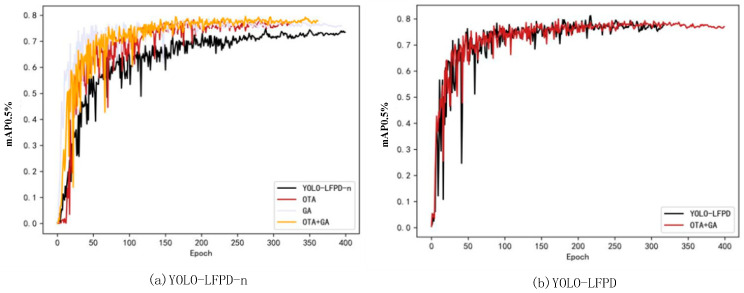
(**a**) is the experimental diagram of the YOLO-LFPD-n parameter, and (**b**) is the experimental diagram of the YOLO-LFPD parameter.

**Figure 14 biomimetics-09-00607-f014:**
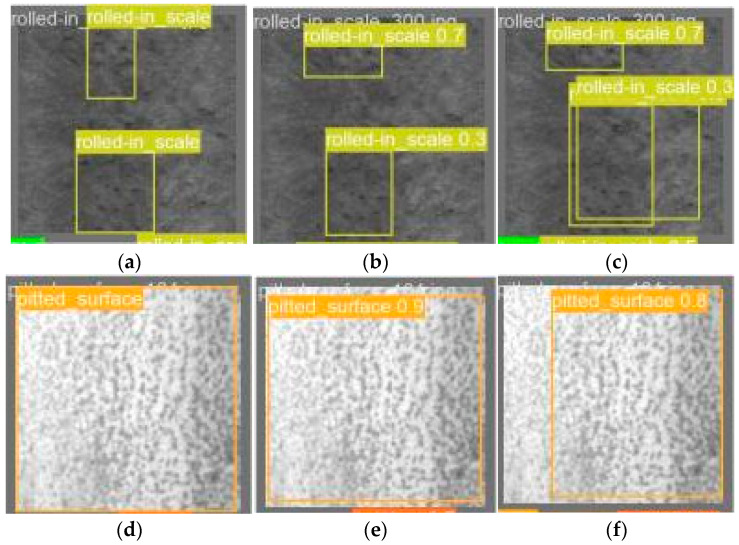
Shows a comparison of the detection frames for the NEU dataset. The original labels are displayed in (**a**,**d**), while (**b**,**e**) and (**c**,**f**) show the detection frames for YOLO-LFPD and YOLO-FPD, respectively.

**Figure 15 biomimetics-09-00607-f015:**
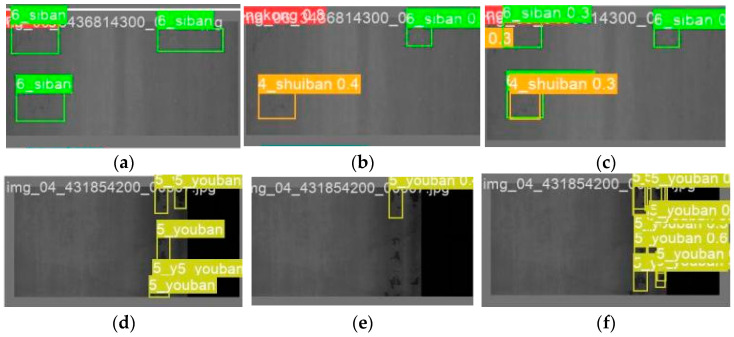
Shows a comparison of the detection frames for the GC10 dataset. The original labels are represented by (**a**,**d**), while (**b**,**e**) and (**c**,**f**) represent YOLOv5 and YOLO-LFPD, respectively.

**Figure 16 biomimetics-09-00607-f016:**
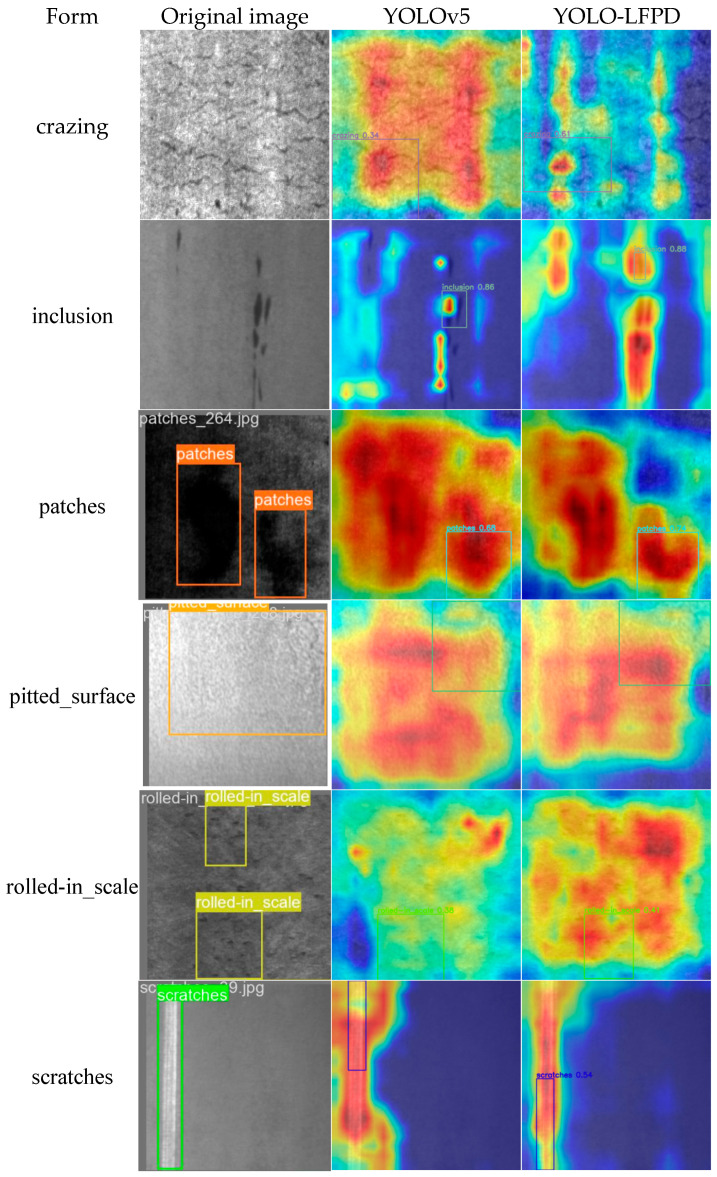
Shows a heat map comparison of the NEU datasets. The left panel displays the raw labels, while the middle and right panels show the results obtained using YOLOv5 and YOLO-LFPD, respectively.

**Table 1 biomimetics-09-00607-t001:** PKU-Market-PCB Pictures.

Typology	Training Set	Validation Set	Test Set	Subtotal
leakage holes	76	23	16	115
open circuits	79	24	6	109
rat bites	86	27	9	122
short circuits	87	16	13	116
stray copper	79	26	10	115
stray	78	22	16	116

**Table 2 biomimetics-09-00607-t002:** Experimental environment.

Experimental Environment	Configuration Parameters
CPU	Intel(R)i5-12400
GPU	NVIDIA 3060 (12 GB)
Deep Learning Frameworks	Pytorch3.8
Programming Language	Python3.7
GPU Acceleration Library	CUDA11.2 CUDNN11.2

**Table 3 biomimetics-09-00607-t003:** YOLO-FPD comparison experimental results on NEU-DET.

Model	mAP@0.5	AP(%)						Model Parameter
		Cr	In	Ps	Pa	Rs	Sc	
SDD [39]	67.1	38.2	72.1	72.0	87.9	65.3	66.9	93.2 M params, 116.3 GFLOPs
Faster R-CNN	71.3	37.6	80.2	81.5	85.3	54.0	89.2	-
ES-Net [39]	79.1	60.9	82.5	95.8	94.3	67.2	74.1	148.0 M params
LDD-Net [37]	74.4	−	−	−	−	−	−	21.5 GFLOPs
Efficiendet [39]	70.1	45.9	62.0	85.5	83.5	70.7	73.1	199.4 M params, 12.6 GFLOPs
DEA-RetinaNet [39]	79.1	60.1	82.5	95.8	94.3	67.2	74.1	168.8 M params
DCC-CenterNet [39]	79.4	45.7	90.6	82.5	85.1	76.8	95.8	131.2 M params
YOLOv3 [39]	69.9	28.1	74.6	78.7	91.6	54.1	92.5	236.3 M params, 33.1 GFLOPs
TAMD [40]	77.9	56.8	82.8	82.6	92.0	60.5	92.4	−
TD-Net [34]	76.8	−	−	−	−	−	−	−
YOLOv5n	76.0	40.1	87.3	82.7	90.4	64.0	91.4	280 layer, 3 M params, 4.3 GFLOPs, 6.7 MB
YOLOv5s	77.3	46.1	82.2	87.8	91.1	64.9	91.8	280 layers, 12.3 M params, 16.2 GFLOPs, 25.2 MB
YOLOv7-Tiny	72.4	37.0	82.8	82.3	87.8	55.5	89.0	263 layers, 6 M params, 13.2 GFLOPs, 12.3 MB
YOLOv7	73.4	36.8	85.6	80.7	88.1	58.7	90.4	415 layers, 37.2 M params, 104.8 GFLOPs, 74.9 MB
DEA-RetinaNet [39]	79.1	60.1	82.5	95.8	94.3	67.2	74.1	168.8 M params
DCC-CenterNet [39]	79.4	45.7	90.6	82.5	85.1	76.8	95.8	131.2 M params
YOLOv3 [39]	69.9	28.1	74.6	78.7	91.6	54.1	92.5	236.3 M params, 33.1 GFLOPs
YOLOX [41]	77.1	46.6	83.1	83.5	88.6	64.8	95.7	– 9 M params, 26.8 GFLOPs, 71.8 MB
LFF-YOLO [39]	79.23	45.1	85.5	86.3	94.5	67.8	96.1	– 6.85 M params, 60.5 GFLOPs, –
MSFT-YOLO [42]	75.2	56.9	80.8	82.1	93.5	52.7	83.5	–
YOLO-V3-based model [43]	72.2	–	–	–	–	–	–	–
ST-YOLO [41]	80.3	54.6	83.0	84.7	89.2	73.2	97.0	55.8 M params,
RDD-YOLO [44]	81.1	–	–	–	–	–	–	–
YOLOv8s	74.7	43.6	82.2	78.1	94.0	66.8	83.3	225 layers, 11.1 M params, 28.4 GFLOPs, 21.5 MB
WFRE-YOLOv8s [45]	79.4	60.0	81.4	82.5	93.8	73.8	84.8	13.8 M params, 32.6 GFLOPs
YOLO-FPD	79.6	56.2	83.0	88.4	87.5	69.4	93.2	232 layers, 13.2 M params, 20.2 GFLOPs, 25.4 MB
YOLO-LFPD-n	78.3	50.4	81.7	86.7	89.7	72.4	89.2	238 layers, 1.6 M params, 3.8 GFLOPs, 3.3 MB
YOLO-LFPD	81.2	63.0	82.4	89.8	86.5	71.9	93.9	238 layers, 6.4 M params, 14.1 GFLOPs, 12.5 MB

**Table 4 biomimetics-09-00607-t004:** YOLO-LFPD comparison experiment results on GC10-DET dataset.

Model	mAP@0.5	Model Parameter
YOLOv5s	65.5	280 layers, 12.3 M params, 16.2 GFLOPs, 25.2 MB
LSD-YOLOv5 [16]	67.9	2.7 M params, 9.1 GFLOPs
MFAM-Net [46]	66.7	−
TD-Net [34]	71.5	−
Improved YOLOX [21]	70.5	−
WFRE-YOLOv8s [45]	69.4	13.8 M params, 32.6 GFLOPs
YOLO-LFPD	72.8	238 layers, 6.4 M params, 14.1 GFLOPs, 12.5 MB

**Table 5 biomimetics-09-00607-t005:** Comparison of experimental results of YOLOv5s-FPD on VOC2007.

Model	mAP@0.5	Model Parameter
Tiny RetinaNet [47]	70.0	−
EfficientDet [47]	69.0	−
TDD-Net [48]	95.1	−
YOLOX [47]	92.3	9 M params, 26.8 GFLOPs, 71.8 MB
YOLOv5s	94.7	280 layers, 12.3 M params, 16.2 GFLOPs, 25.2 MB
YOLOv7 [48]	95.3	415 layers, 37.2 M params, 104.8 GFLOPs, 74.9 MB
YOLO-MBBi [48]	95.3	−
PCB-YOLO [47]	96.0	−
DInPNet [49]	95.5	−
TD-Net [34]	96.2	−
YOLO-LFPD	98.2	238 layers, 6.4 M params, 14.1 GFLOPs, 12.5 MB

**Table 6 biomimetics-09-00607-t006:** Comparative experiments between FasterNet and RepVGG on NEU dataset.

Model	mAP@0.5	Params	GFLOPs
YOLO-FPD	79.6	13.2 M	20.2
YOLO-FPD + FasterNet	79.8	8.4 M	14.9
YOLO-FPD + RepVGG	80.4	11.2 M	19.4
YOLO-LFPD	81.2	6.4 M	14.1

**Table 7 biomimetics-09-00607-t007:** Shows the results of the experiment comparing the inference speed of models on the NEU dataset.

Model	mAP@0.5	Parameter/M	GFLOPs	Speed of Reasoning/ms
YOLOv5n	76.0	3	4.3	3.8
YOLOv5s	77.3	12.3	16.2	7.0
YOLOv7-Tiny	72.4	6	13.2	9.0
YOLOv7	73.4	37.2	104.8	9.5
YOLOv8s	74.7	11.1	28.4	7.5
YOLO-FPD	79.6	13.2	20.2	7.2
YOLO-LFPD-n	78.3	1.6 (53%)	3.8 (88%)	3.2 (84%)
YOLO-LFPD	81.2	6.4 (52%)	14.1 (87%)	5.4 (77%)

**Table 8 biomimetics-09-00607-t008:** FasterNet module fusion location results.

Fusion Position	mAP@0.5	FPS
Backbone Network	79.8	139
Neck Network	79.5	137
Detection Heads	79.3	136

**Table 9 biomimetics-09-00607-t009:** Shows the results of the parameter experiment conducted on the NEU dataset.

Model	mAP@0.5	mAP@0.95
YOLO-LFPD-n	74.0	40.9
OTA + YOLO-LFPD-n	77.6	42.5
GA + YOLO-LFPD-n	77.4	43.0
OTA + GA + YOLO-LFPD-n	78.3	43.2
OTA + GA + YOLO-LFPD	80.5	44.4
YOLO-LFPD	81.2	44.3

**Table 10 biomimetics-09-00607-t010:** Shows the results of the pruning experiments conducted on the NEU dataset.

Pruning Method	GFLOPs	Model Size/MB	Parameter/M	mAP@0.5
YOLO-LFPD	3.5	12.5	6.4	81.2
L1-1.5×	2.6	8.5	4.3	77.3
L1-2.0×	1.72	6.4	3.17	75.6
SLIM-1.5×	2.6	8.5	12.3	68.4
SLIM-2.0×	1.71	6.4	3.16	64.5
SLIM-4.0×	0.88	3.3	1.66	61.4
LAMP-1.5×	2.6	8.5	4.31	71.8
LAMP-2.0×	1.72	6.4	3.17	72.9
LAMP-4.0×	0.88	3.3	1.66	70.4
Group_slim-1.5×	2.6	8.5	4.3	74.0
Group_slim-2.0×	1.72	6.4	3.16	70.6
Group_taylor-1.5×	2.6	8.5	4.3	73.1
Group_taylor-2.0×	1.72	6.4	3.16	71.4
Growing_reg1.2×	2.91	10.7	5.37	74.1
Growing_reg1.5×	2.6	8.5	4.3	74.7
Group_norm1.2×	2.9	10.4	5.37	75.8
Group_norm1.4×	2.5	8.9	4.56	77.1
Group_norm1.5×	2.6	8.5	4.31	77.4
Group_norm-2.0×	1.72	6.4	3.16	75.3

## Data Availability

The data that support the findings of this study are available in [NEU-DET], [GC10-DET] and [PKU-Market-PCB] at [http://faculty.neu.edu.cn/songkc/en/zhym/263264/list/index.htm (accessed on 5 September 2024)], [https://aistudio.baidu.com/datasetdetail/90446] and [https://aistudio.baidu.com/datasetdetail/211346 (accessed on 5 September 2024)].

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
