# Peer review of "YOLO-LFPD: A Lightweight Method for Strip Surface Defect Detection"

_biomimetics, 2024, doi:10.3390/biomimetics9100607_

Round 1

Reviewer 1 Report

Comments and Suggestions for Authors

In the Article «YOLO-LFPD: A Lightweight Method for Strip Surface Defect Detection» by authors: Jianbo Lu, Mingrui Zhu, KaiXian Qin, and XiaoYa Ma, another method for detecting defects on the surface of steel based on deep learning is presented. Compared with other methods, it has some advantages. This is a slight increase in accuracy and inference speed.

As comments, I would like to note some difficulties in perceiving the text, for example:

1. It is not entirely clear why in the experiment sets the epoch of each model to 400, (lines 458-459) and not more or less? Of course, one can guess, but it would be better if the authors explained the reason for choosing the parameters.

2. It is also unclear why the initial learning rate is 0.1, which then decreases to 0? This is line 459. After all, decreasing the speed to zero means stopping the work? Then the performance (productivity) becomes zero.

3. The expression "Although the accuracy of YOLOv5 improves rapidly in the early stage of training, the accuracy cannot be further improved due to the overfitting situation of the model, while YOLO-LFPD reduces the depth of the model and increases the residual structure compared to YOLO-FPD, and defective features can be better noticed under the condition of an insufficient number of samples, and it turns to reach the top when it is trained up to about 250 epochs, whereas YOLO-FPD reaches the top at nearly 400 epochs." (lines 499-505) is very difficult to read. It is better to break it down into several simple sentences.

But in general, according to the results presented in tables 3-7, it is clear that the YOLO-LFPD model proposed by the authors has some advantages compared to other models for determining strip steel surface defect recognition research. I cannot say how important the 3% increase in accuracy obtained by the authors is in industrial production, but the authors have obtained it. This result should be further evaluated by steel industry experts. I believe that the article “YOLO-LFPD: A Lightweight Method for Strip Surface Defect Detection” presented by the authors: Jianbo Lu, Mingrui Zhu, KaiXian Qin, and XiaoYa Ma meets the requirements of the journal “Biomimetics” and can be published.

Author Response

Thank you very much for your feedback on our article and for your valuable comments, to which we make the following relevant responses.Paper revisions and additions that we have made in the Revision Trace Draft are highlighted in yellow within the draft.

  1. It is not entirely clear why in the experiment sets the epoch of each model to 400, (lines 458-459) and not more or less? Of course, one can guess, but it would be better if the authors explained the reason for choosing the parameters.

Thank you for pointing out this problem, the reason for setting the epoch to 400 has something to do with experimental experience. In our experiments, we set a parameter to automatically end the training, for example, 100, during training, it will record which epoch was the last highest accuracy, and then end the experiments early when there is no higher accuracy after 100 epochs in the next training. In our experiments, the YOLO-LFPD model usually reaches its peak at 250 epochs, while the variants of YOLO-LFPD and other models tend to reach their peaks at no more than 400 epochs, so we set the epoch to 400 in order to make the experimental environment as consistent as possible. we will add the following to the original text :

The performance of these models is compared to other models. The experiment sets the epoch of each model to 400, the initial learning rate to 0.1, which then decreases to 0. The optimizer uses SGD.400 epoch ensures that all models involved in the comparison are trained to the maximum extent possible.The GC10-DET dataset has a high image resolution, but due to limitations in the experimental device's video memory, the training batch size is set to 16 for both training and testing.

  1. It is also unclear why the initial learning rate is 0.1, which then decreases to 0? This is line 459. After all, decreasing the speed to zero means stopping the work? Then the performance (productivity) becomes zero.

Thanks for pointing this out, it was an oversight on our part. This description may be inaccurate, as the learning rate already tends towards 0 during training. lr0 and lrf are parameters related to the learning rate, and lrf is usually multiplied by the initial learning rate, lr0, to determine the learning rate for each subsequent epoch. Therefore, we will modify this description:

The performance of these models is compared to other models. The experiment sets the epoch of each model to 400, with the initial learning rate set to 0.1 and then gradually converging to 0. 400 epoch ensures that all models involved in the comparison are trained to the maximum extent possible.The optimizer uses SGD.

  1. The expression "Although the accuracy of YOLOv5 improves rapidly in the early stage of training, the accuracy cannot be further improved due to the overfitting situation of the model, while YOLO-LFPD reduces the depth of the model and increases the residual structure compared to YOLO-FPD, and defective features can be better noticed under the condition of an insufficient number of samples, and it turns to reach the top when it is trained up to about 250 epochs, whereas YOLO-FPD reaches the top at nearly 400 epochs." (lines 499-505) is very difficult to read. It is better to break it down into several simple sentences.

Thank you for pointing this out, we will be revising this section as follows:

Although the accuracy of YOLOv5 improves rapidly in the early stages of training, the accuracy cannot be improved further due to the overfitting situation of the model, peaking at nearly 400 epochs. YOLO-LFPD, on the other hand, reduces the depth of the model and increases the residual structure compared to YOLO-FPD, which allows better attention to defective features with insufficient number of samples, and peaks when training reaches about 250 iterations, which suggests that the optimised structure captures the target features better.

We hope that our revisions and responses address your concerns and strengthen the manuscript. We are grateful for the opportunity to improve our work.

Reviewer 2 Report

Comments and Suggestions for Authors

The paper is appropriate for the journal.

THe subject is of interest for speialists.

In order to improve the papers and make it for a wider readers some comments are necessary.

The authors present in this paper a tutorial on the methods to detect imperfections. Mainly the attention has been devoted to the today well used Convolutional Neural Networks. The papers contains also new results both as regards the methods ans as regard the applications. It is addressed to a specializd reader. The authors must improve both the introduction and the conclusions. The authors could propose also other approaches like that os Cellular Nonlinear Networks for achiving good results in the studied topic. A remark about this possible approach must be reported and in particular they must to keep into account the following contribution;

Self-organization in a two-layer CNN Authors Paolo Arena, Salvatore Baglio, Luigi Fortuna, Gabriele Manganaro Pub Date:1998/2 Pubblicazione IEEE Transactions on Circuits and Systems I: Fundamental Theory and Applications Vol 45 Number2 Pages 157-162 Editor  IEEE   The authors must concentrate their omments during the discussion.

Author Response

Dear Esteemed Reviewer,

From the depths of my heart, I extend my gratitude for your painstaking review of my paper and the invaluable insights shared amidst your demanding schedule. Concerning the cellular nonlinear network (CNN) you highlighted, I'm profoundly appreciative of its theoretical advancements and application potentials; however, within the domain of defect detection, its practical utility remains unproven. Allow me to elucidate:

  • The Pivotal Role of YOLO: YOLO (You Only Look Once), renowned as a real-time target detection framework, has carved out a significant niche in the realm of image recognition through its unparalleled speed and precision. By processing entire images at once, YOLO predicts bounding boxes and class probabilities effectively. Its end-to-end learning framework ensures swift detection speeds and high accuracy, making it exceptionally suitable for real-time defect detection tasks, particularly in industries reliant on automation and stringent quality controls.

  • Alternate Approaches to Image Recognition: While convolutional neural networks (CNNs) and region-based CNNs (R-CNNs) were not extensively discussed, their foundational roles in advancing image recognition are irrefutable. In the interest of maintaining thematic coherence and concise presentation, YOLO was prioritized for detailed analysis; nonetheless, I acknowledge the significance and research value of alternative methodologies.

  • Challenges Faced by Cellular Nonlinear Networks in Defect Detection: Despite their prowess in modeling intricate biological phenomena, cellular nonlinear networks exhibit limitations in the defect detection arena. Primarily, defect detection entails analyzing diverse forms of image and sensor data, markedly divergent from biochemical cellular processes. Their inherent complexity hinders efficient feature extraction pivotal for rapid image recognition. Moreover, they lack adaptability essential for detecting varied defect types under differing scenarios.

In essence, the exclusion of cellular nonlinear networks from this paper’s discourse stems from a focused commitment to YOLO’s advancement within the context of image recognition. Simultaneously, I concur with the imperative of interdisciplinary convergence, vowing to maintain vigilance over potential synergies between these domains in forthcoming investigations.

Your guidance has been duly noted, rest assured that your recommendations shall receive serious consideration. In upcoming studies, I intend to probe further into algorithms tailored for defect detection, aspiring to contribute significantly towards the discipline’s progression.

Await my future endeavors with anticipation. Your unwavering support and encouragement serve as my compass, guiding every stride forward.
